# Evaluation of post-operative complications, outcome, and long-term owner satisfaction of elbow arthrodesis (EA) in 22 dogs

Elaine V. Dinwiddie[1], Aaron Rendahl[2], Stan Veytsman[3], Guillaume Ragetly[4], Albert C. Lynch[1], Brianna Miniter[5], Ron Ben-Amotz[6]*

1 Small Animal Surgery Department, Veterinary Specialty and Emergency Center, Levittown, PA, United States of America, 2 Statistics and Informatics Department, College of Veterinary Medicine at the University of Minnesota, St. Paul, MN, United States of America, 3 Small Animal Surgery Department, College of Veterinary Medicine at the University of Minnesota, St. Paul, MN, United States of America, 4 Small Animal Surgery Department, Centre Hospitalier Veterinaire Fregis, Arcueil, France, 5 Small Animal Surgery Department, BluePearl Pet Hospital, Clearwater, FL, United States of America, 6 Small Animal Orthopedics Department, Koret School of Veterinary Medicine at The Hebrew University of Jerusalem, Rehovot, Israel

* ron.benamotz@mail.huji.ac.il

**Data Availability Statement:** The minimal data set is listed in the paper as Table 1.

## Abstract

The objective of this study was to report post-operative complications and outcomes in canines undergoing elbow arthrodesis (EA) with fixation techniques including bone plate fixation with a non-locking dynamic compression plate (DCP), bone plate fixation with a locking plate (LCP), and external skeletal fixator (ESF). Medical records of twenty-two cases that underwent EA between January 2009-December 2019 from 8 referral hospitals including both private practice and academic institutions were reviewed. Post-operative complications were classified as either minor or major, surgical evaluations were performed 8 weeks post operatively, and a follow-up questionnaire was sent to owners. Of the total 22 cases that met inclusion criteria, a total of 19/22 cases had complications, 12 major and 7 minor. Complications reported in 8/9, 7/9, and 4/4, for the DCP, LCP, and ESF fixation groups, respectively. Mild to moderate mechanical lameness was identified at surgical evaluation in 16/22 cases. Complete radiographic bone healing was achieved after 9 weeks in 19/22 cases. Long term owner follow up was available in 14/22 cases. Owners reported a good to normal quality of life in 13/14 cases and poor in one case. The majority of owners (11/14) reported good to excellent satisfaction with the outcome irrespective of persistent lameness. This study demonstrates that successful EA can be achieved using a variety of fixation methods, but persistent lameness is expected and complication rate is high.

## Introduction

Lameness associated with the elbow joint is common in dogs and may be congenital, developmental, or traumatic in origin. For many dogs, acceptable limb function may be achieved with surgical and medical therapy; however, in a subset of cases, this is not attained. These cases experience a lower quality of life due to their end-stage elbow joint disease and owners are often faced with the decision between more radical surgical intervention, or euthanasia.

**Funding:** The author(s) receive no specific funding for this work.

**Competing interests:** The authors have declared that no competing interests exist.

The goals of surgical intervention for end-stage elbow pathology include stabilization of the joint, alleviation of pain, and maintaining limb functionality. There have been various procedures reported to achieve this, including various ostectomy techniques, elbow prosthetic arthroplasty, joint resurfacing, joint denervation, and arthrodesis [1–9]. Currently, an effective elbow joint replacement system is unavailable, and elbow arthrodesis (EA) may be the only available salvage procedure for cases where amputation is not a viable option.

Elbow arthrodesis is an infrequently reported procedure with only 6 cases reported in the past 10 years of literature [10, 11]. In a paper published in 1996, EA was reported in twelve dogs, and it was concluded that while acceptable function was attained, mechanical lameness persisted [10]. Post-operatively, mechanical lameness can be anticipated due to the fixed angle of the elbow joint [10]. Unfortunately, literature reporting major and minor post-operative complications in dogs having undergone EA is lacking.

There are currently pre-contoured arthrodesis plates available for the carpus and tarsus. A locking implant pre-contoured for medial elbow arthrodesis has been developed in the last 2 years and used successfully in a previous study [11]. However, there are currently no commercially available implants dedicated for the use of caudal EA application, the development of which may improve post-operative outcomes in cases.

The objective of this study was to report post-operative complications, long term outcome, and owner satisfaction in dogs following EA stabilized by dynamic compression plates (DCP), locking plates (LCP), and external skeletal fixation (ESF).

## Materials and methods

Medical records from 8 different referral institutions were reviewed from January 2009 to December 2019. Dogs having undergone EA were included and grouped based on fixation method. Details regarding age, gender, bodyweight, indication for EA, surgical technique, angle of elbow fixation, post-operative complications (major and minor), post-operative lameness exams, and radiographic outcome were obtained from medical records. Surgical complications were recorded for each type of fixation and divided into two groups: minor complications and major complications. Minor complications were defined as those that resolved with non-surgical intervention, such as superficial surgical site infections (SSI), single screw/pin breakage or loosening, and superficial bandage complications [11, 12]. Major complications were those that required revision surgery and/or hospitalization such as deep SSI, bone fracture, and bone plate fracture [12].

The intended arthrodesis angle was based on previous reports of standing elbow angles in dogs ranging from 110–159 degrees [8–10, 13–16]. The angle of elbow fixation was measured on immediate post-operative radiographs in all cases. It was measured as the intersection of the mechanical axis of the distal humeral diaphysis and mechanical axis of the proximal radius [8, 10].

Outcome was assessed by post-operative surgical evaluations and radiographic imaging that were performed at an average of 9 weeks post operatively. Lameness exams were categorized as none, mild, moderate, or severe at 8-week post-operative surgical evaluations.

Long-term follow-up was completed via telephone questionnaire with owners at the time of data collection. The questionnaire was a modified version of a previously reported owner-questionnaire used to assess the dog's function and owner-perceived outcome [10]. Owners were asked to report their pet's current overall lameness as either none (no detectable lameness), mild (intermittent weight bearing lameness), moderate (frequent weight bearing lameness), or severe (non-weight bearing lameness). In addition, owners reported the current quality of life of their pet as either normal, good, fair, or poor. The overall owner satisfaction with the procedure was reported as excellent good, fair, or poor.

## Results

A total of 21 canines met the inclusion criteria including one case with bilateral staged EA procedures, resulting in 22 total cases. Of the cases included, there were 8 castrated males, 4 intact males, 5 spayed females, and 4 intact females. Median age at the time of surgery was 4.0 years (range, 4 mo. -10 yrs.). Median body weight was 17.8kg [range, 1.25kg-42kg]. Breeds included were Labrador Retrievers (4), German Shepherd dogs (3), Bull Mastiff (1), Golden Retriever (1), Samoyed (1), Coton de Tulear (1), Yorkshire Terrier (3), Cavalier King Charles Spaniel (1), Cocker Spaniel (1), Miniature Pinscher (1), Rat Terrier (1), French Bulldog (1), and mixed breeds (3).

The etiology of the end stage joint disease necessitating EA were as follows: distal humeral fracture failure, non-union, or mal union (7), luxation or subluxation (5), chronic infection of humeral condylar fracture (1), end-stage elbow osteoarthritis (7), pathologic condylar fracture (1), and failure of elbow arthroplasty (1). Seven of the 22 cases had concurrent orthopedic disease including degenerative joint disease of the carpi, stifles, and hips, medial patella luxation correction, femoral head and neck ostectomy, cranial cruciate ligament rupture repairs (tibial plateau leveling osteotomy, tibial tuberosity advancement), and hip dysplasia.

### Surgical procedure

Surgical procedures were performed by, or under the direct supervision of, diplomates of the American and/or European College of Veterinary Surgeons. Implants were selected at the discretion of the surgeon and included various manufacturers. In total, twenty-two EA procedures were included in this study, and were stabilized by ESF (4/22), DCP (9/22) or LCP (9/22) fixation. Bone plates were applied medially (1/18), laterally (3/18), or caudally (14/18). A 3.5 mm DCP was used in all procedures, and a 2.0/2.4/3.5 mm LCP was used in all procedures. A lateral 2.0 LCP was placed to augment a caudal LCP in one case. Prior to plate fixation, transarticular screws and/or pins were used in 14/18 cases. ESF configurations used included a Type 1A configuration, a Type 1B configuration, a Type 1A with a tie-in configuration, and a Type 1A configuration augmented with Kirschner and cerclage wire.

An olecranon osteotomy was performed in 21/22 procedures to provide access to articular cartilage that was removed with either a bur attached to a high-speed air drill (CONMED, Utica, NY, United States) or with a handheld curette. Once completed, bone graft was placed in the humeroulnar, humeroradial, and radioulnar joints in 17/22 cases. Of the 17 procedures that were grafted, an autogenous bone graft was used in 15 cases. A mixture of autogenous bone and allograft bone was used in one case and bone putty (Evergraft, Everost Inc., Sturbridge, MA, United States) was used in one case. Harvest sites used were the humerus, radius, ulna, ilium as well as bone removed from the surgical site to facilitate reduction. Documentation of graft source was unavailable in one case.

Three cases had chronic elbow luxation that required additional procedures including distal humeral, ulnar, and/or radial osteotomies to achieve appropriate mechanical angulation and alignment. These procedures along with the olecranon osteotomies, were stabilized with either pins, screws, pin and screw, or pin and tension band prior to application of the definitive repair.

One intra-operative complication was reported, an iatrogenic humeral diaphyseal fissure that occurred during application of a caudal LCP but did not require revision.

### Imaging

No major complications were identified on immediate post-operative radiographs. The mean angle of fixation measured on post-operative radiographs was 120.9 degrees (range 93.5–140 degrees).

Bone healing at the EA site was identified as either progressing or complete on average 9 weeks (range 8–18 weeks) post-operatively in 19/22 cases.

Five cases did not receive a bone graft. Three of those cases had appropriate bone healing identified on 8-week post-operative radiographs. One dog was euthanized due to recurrence of soft tissue sarcoma prior to follow up radiographs. One case resulted in EA nonunion and pin migration that was identified on 8-week post-operative radiographs. The dog subsequently underwent ESF revision with bone screws and wire. Bone graft was placed during the EA revision and bone fusion was identified on radiographs 18 weeks post-operative from the initial surgery.

**DCP complications and outcome.** Of the 9 reported DCP cases used for EA, complications occurred in 8/9. Of those, five cases had major complications and 3 cases had minor complications (Table 1). Minor complications included SSI, limb swelling, seroma, pain, superficial sores, and broken olecranon screw. Major complications included implant migration, implant fracture, arthrodesis nonunion, and deep SSI. Major complications resulted in partial explantation in 1/4 cases and complete explantation in 3/4 cases. One case (case #5) had 3 pins explanted 9 weeks post operatively due to pin migration while the primary construct remained intact (Fig 1). A methicillin-resistant *Staphylococcus pseudintermedius* (MRSP) infection developed in one case (case #6) that led to implant exposure, non-healing wounds, and nonunion. Revision surgery versus amputation was discussed, but the owners elected medical management with serial bandage changes. The dog was humanely euthanized 7 months post operatively due to progressive lameness and decreased quality of life. Bilateral EA procedures were performed in one case, staged two months apart (case #7, #8). Complete explantation was performed on the right elbow post operatively due to screw loosening and bone plate shifting. The left side resulted in EA failure due to shifting and fracture of the implant 7 weeks post-operatively and was revised with a caudal DCP fixation.

Surgical evaluations were performed 8 weeks post operatively in all DCP cases. Seven of the 9 cases had mild to moderate mechanical lameness with varying degrees of circumduction. Two cases had severe non weight bearing lameness. Seven cases had complete bone healing identified on radiographs at a median time of 9 weeks post-operatively (range 8–18 weeks). One case (case #3) did not have follow-up radiographs performed and nonunion was identified in one case (case #6).

Long-term follow-up was completed via telephone questionnaire in 7/9 DCP cases (range of 7 months-9 years post-operatively) (Table 2). Owners reported no lameness in one case, mild to moderate lameness in 3/7 cases, and severe lameness in 3/7 cases. A fair to good quality of life was reported in 6/7 cases with an overall good to excellent owner satisfaction in 6/7 cases. Owners of the dog with staged bilateral EAs (case #7, #8) reported severe non weight bearing lameness of both limbs but good quality of life. However, due to persistent bilateral thoracic limb lameness and concurrent bilateral pelvic limb orthopedic disease, the dog was placed in a mobility cart for the thoracic limbs and a wheelchair for the pelvic limbs. One owner reported poor quality of life of the dog and poor satisfaction with the procedure (case #6).

**LCP complications and outcome.** Of the 9 reported LCP cases used for EA, complications occurred in 7/9 cases. Of those, 3 cases had major complications and 4 cases had minor complications (Table 1). Minor complications included SSI, transient neuropraxia, broken olecranon screw, and humeral fissure (Fig 2). Major complications included screw fracture, screw loosening, deep SSI, and radius/ulna fracture. Superficial SSI occurred in one case that resulted in dehiscence of the surgical site (case #10), and at 9 weeks post-operatively, a fracture of the radius and ulna occurred distal to the implant. The fracture was repaired with a cranially applied 3.5mm LCP and the dehisced surgical arthrodesis site was debrided and closed.

**Table 1. Individual case post-operative minor and major complications.**

| No. | Signalment | Fixation | Minor complications | Major Complications |
|---|---|---|---|---|
| 1 | Bullmastiff 7 YO, FS, 42 kgs | DCP | Broken olecranon screw | Deep SSI |
| 2 | Cocker Spaniel 6 YO, CM, 12 kgs | DCP | Pain | None |
| 3 | Labrador Retriever 9.5 YO, FS, 33 kgs | DCP | Pain, limb swelling, transient radial neuropraxia | None |
| 4 | Labrador Retriever 8 YO, M, 34 kgs | DCP | SSI | None |
| 5 | German Shepherd 1.5 YO, SF, 39.4 kgs | DCP | SSI, incisional erythema | Pin migration |
| 6 | Samoyed 3.5 YO, CM, 29.9 kgs | DCP | SSI, pain, sores, bruising, bone plate exposure, muscle atrophy | Deep SSI, arthrodesis nonunion |
| 7 | Labrador Retriever 3 YO, CM, 23 kgs | DCP (right) | SSI, pain, limb swelling | Screw loosening, bone plate shifting |
| 8 | Labrador Retriever 3 YO, CM, 23 kgs | DCP (left) | Limb swelling, seroma formation | Deep SSI, bone plate fracture, implant shifting |
| 9 | Golden retriever 10 YO, SF, 26 kgs | DCP | None | None |
| 10 | German Shepherd 3 YO, CM, 34.4 kgs | LCP | SSI | Deep SSI, incision dehiscence, implant exposure, radius/ulna fracture |
| 11 | Mixed breed 7 YO, CM 11.1 kgs | LCP | SSI, nonhealing wound | Proximal screw loosening, pin migration, distal screw fracture |
| 12 | Coton de Tuléar 1.9 YO, SF, 8 kgs | LCP | SSI | None |
| 13 | Yorkshire 7 YO, CM, 4 kgs | LCP | None | None |
| 14 | Mixed breed 0.3 YO, M, 4 kgs | LCP | Broken olecranon screw | None |
| 15 | Yorkshire 8 YO, F, 2.7 kgs | LCP | None | None |
| 16 | Yorkshire 0.6 YO, M, 2.2 kgs | LCP | Humeral fissure | None |
| 17 | Cavalier King Charles Spaniel 0.3 YO, F, 2.8 kgs | LCP | Transient radial neuropraxia, olecranon segment displacement | None |
| 18 | Mixed breed 3 YO, CM, 12 kgs | LCP | None | Deep SSI |
| 19 | Miniature Pinscher 1 YO, F, 1.25 kgs | ESF | Bandage sores, muscle atrophy, osteomyelitis, non-weight bearing lameness | Radius fracture |
| 20 | German Shepherd 1 YO, M, 35.2 kgs | ESF | SSI, pain, limb swelling, bandage sores, pin site inflammation, muscle atrophy, carpal hyperextension | Deep SSI, pin loosening |
| 21 | Rat Terrier 0.25 YO, F, 2.54 kgs | ESF | SSI, pain, self-trauma, pin site inflammation, suture reaction, transient radial neuropraxia, reduced carpal flexion, non-weight bearing lameness | Pin migration, arthrodesis nonunion |
| 22 | French Bulldog 5.5 YO, CM, 14.7 kgs | ESF | SSI, interdigital erythema, limb swelling | Deep SSI, pin loosening, pin fracture |

No. = case number; YO = years old; M = male; F = Female; CM = castrated male; SF = spayed female; kgs = kilograms; DCP = dynamic compression plate;

LCP = locking compression plate; ESF = external skeletal fixator; SSI = surgical site infection.

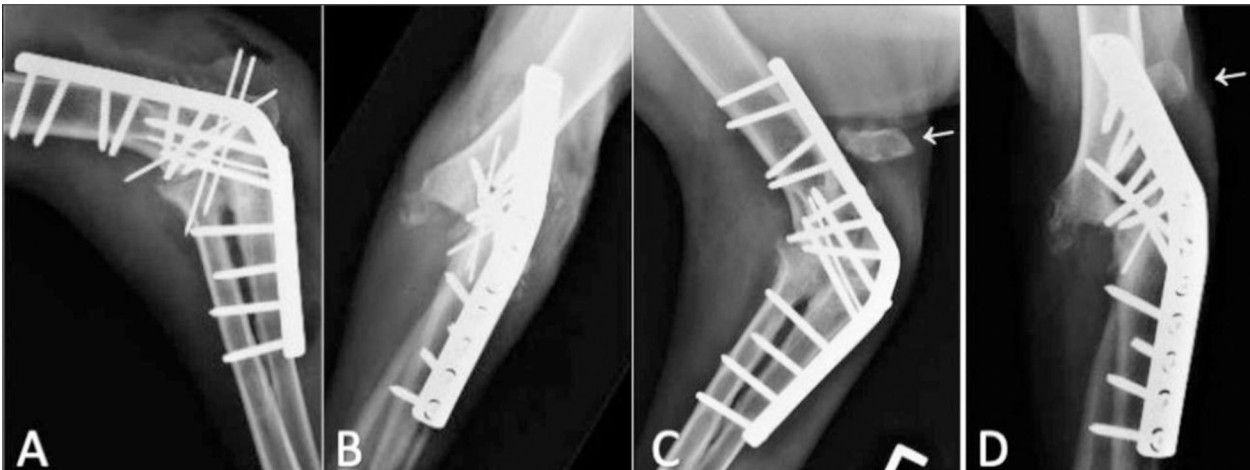

**Fig 1. Post-operative radiographs of DCP EA performed to stabilize a chronic, traumatic elbow luxation (case #5).** (A, B) Three k-wires placed to stabilize the olecranon prior to caudal plate application. One k-wire was placed across the ulna-humerus prior to plate application. (C, D) Pin migration resulted in their removal. This resulted in proximal and lateral olecranon displacement (white arrow).

Surgical evaluations were performed 8 weeks post operatively in all LCP cases. One case had no lameness reported and 8/ 9 cases had mild to moderate mechanical lameness with varying degrees of circumduction. Bone healing was identified on radiographs at a median time of 9 weeks post-operatively in 8 cases.

Long-term follow-up was completed via telephone questionnaire in 4/9 LCP cases (range of 7 months-9 years post-operatively) (Table 2). Owners reported no lameness in one case and mild to moderate lameness in 3/4 cases. Owners reported a good quality of life in 4/4 dogs and good to excellent owner satisfaction in 4/4 cases.

**Table 2. Long term follow-up owner questionnaire regarding dog limb usage, quality of life, and overall owner satisfaction.**

| Owner questionnaire post op DCP elbow arthrodesis in 7 cases | | | |
|---|---|---|---|
| Preoperative lameness severity | None | Mild | Moderate | Severe (7) |
| Current lameness severity at walk | None (1) | Mild (2) | Moderate (3) | Severe (1) |
| Current lameness severity at run | None | Mild (3) | Moderate (1) | Severe (3) |
| Current overall limb disability | None (1) | Mild (1) | Moderate (2) | Severe (3) |
| Current quality of life | Normal (1) | Good (5) | Fair | Poor (1) |
| **Owner questionnaire post op LCP elbow arthrodesis in 4 cases** | | | |
| Preoperative lameness severity | None | Mild | Moderate | Severe (4) |
| Current lameness severity at walk | None (1) | Mild (2) | Moderate (1) | Severe |
| Current lameness severity at run | None (2) | Mild (2) | Moderate | Severe |
| Current overall limb disability | None (1) | Mild (2) | Moderate (1) | Severe |
| Current quality of life | Normal | Good (4) | Fair | Poor |
| **Owner questionnaire post op ESF elbow arthrodesis in 3 cases** | | | |
| Preoperative lameness severity | None | Mild | Moderate | Severe (3) |
| Current lameness severity at walk | None | Mild (1) | Moderate (1) | Severe (1) |
| Current lameness severity at run | None | Mild (2) | Moderate | Severe (1) |
| Current overall limb disability | None | Mild | Moderate (1) | Severe (2) |
| Current quality of life | Normal (2) | Good (1) | Fair | Poor |

Owners' responses to long term follow-up questions for cases are noted in parentheses.

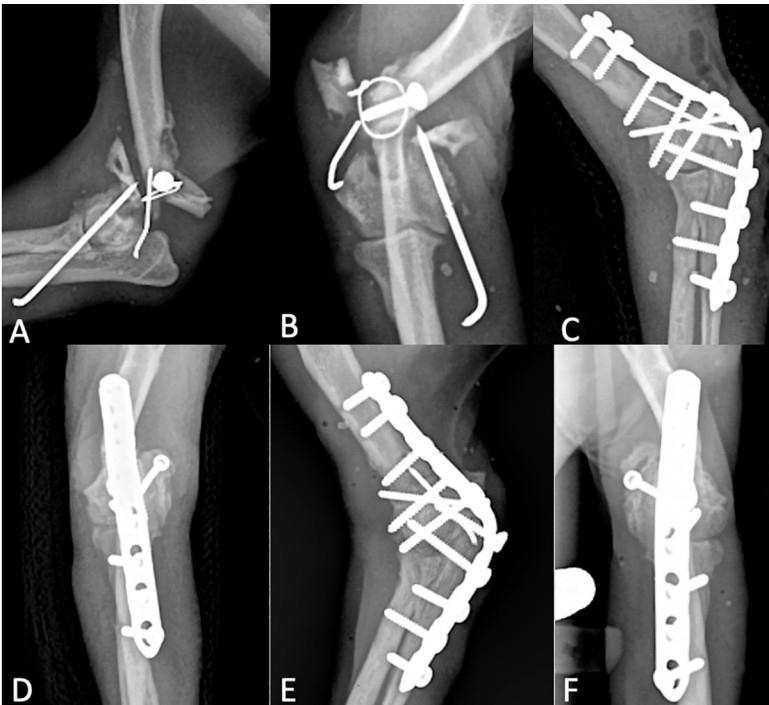

**Fig 2. Radiographs of LCP EA revision for nonunion of previously repaired supracondylar humeral fracture (case #15).** (A, B) Preoperative failed supracondylar humeral fracture repair. (C, D) Revision EA. The previous implants were removed, olecranon osteotomy stabilization with a positional screw. Transarticular pin applied from ulna to humerus prior to caudal LCP application. (E, F) 4-month post-operative healed EA.

**ESF complications and outcome.** Of the 4 reported ESF cases used for EA, complications occurred in 4/4 cases. Both major and minor complications occurred in all cases (Table 1). Minor complications included superficial SSI, pain, non-weight bearing lameness, and muscle atrophy. Major complications included deep SSI, pin loosening, pin breakage, arthrodesis non-union, and long bone fracture.

Two pins were removed due to pain and loosening identified at 8-week post-operative surgical evaluation (case #20). Moderate mechanical lameness was reported at 12-week post-operative surgical evaluation. Bone healing was identified on radiographs at that time and the ESF was removed. The case was lost to long term follow up.

One case had severe, non-weight bearing lameness at 8-week post-operative surgical evaluation (case #19). Radiographs showed complete fusion of the EA site along with a fracture of the radius through the distal pin site. The ESF was removed, and the fracture was managed with a soft padded bandage and caudal splint for 8 weeks. At long term follow up, the owner reported that the dog continued to have severe, non-weight bearing lameness (Table 2). The owner reported a normal quality of life for the dog and fair satisfaction with the outcome. The dog was euthanized 5 years post operatively due to several concurrent pathologies including multiple limb lameness, chronic renal disease, and seizures.

One case had moderate to severe lameness reported at surgical evaluation 8 weeks post-operatively (case #21). Early hypertrophic non-union of the ulnar osteotomy site, loosening of the ESF pins, and cranial intramedullary pin migration were identified on radiographs at that time. The ESF was removed and revised with 3 transarticular cortical screws and a transarticular k-wire. Moderate mechanical lameness was reported at 4-week post-revision surgical evaluation and radiographs showed appropriate fusion of the EA site 18 weeks post-operatively. At

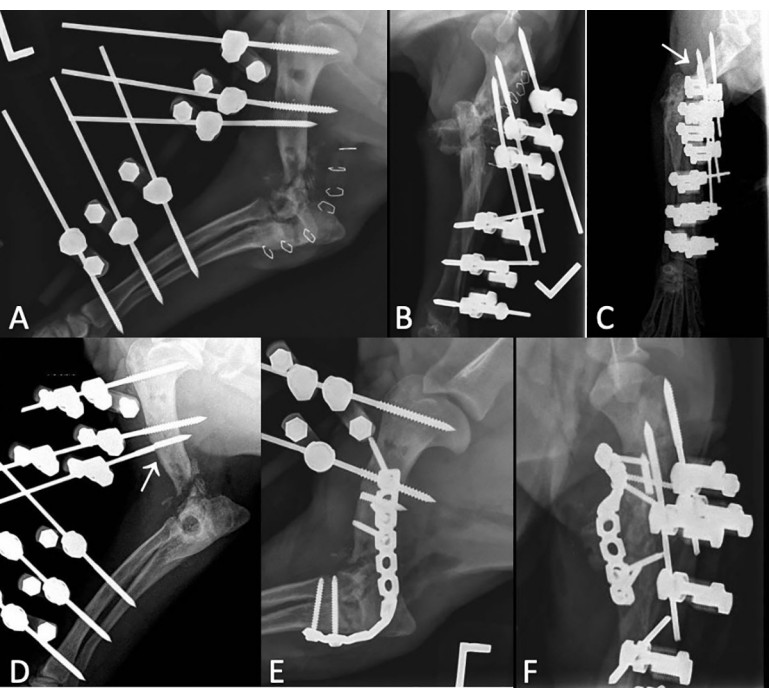

**Fig 3. Radiographs of ESF revision of previously failed arthrodesis of a medial condylar fracture (case #22).** (A, B) Post-operative revision EA procedure with a type 1A ESF. The 3 proximal humeral pins and the 3 distal radial pins were stabilized with a carbon bar. Bone graft was placed. (C, D) 6 weeks post-operative radiographs. Fractured distal humeral ESF pin (white arrow). (E, F) The fractured pin was removed. A caudal reconstruction plate was applied with 3 proximal screws placed in the humerus and 2 distal screws placed through the ulna and radius.

long term owner follow up, the owner reported that the dog had severe lameness. The owner also reported a normal quality of life and fair satisfaction with the outcome.

Acute, non-weight bearing lameness occurred 6 weeks post operatively in one case (case #22). Radiographs showed that the distal humeral ESF pin had fractured. The pin was removed and a 11 hole 2.7mm non-locking reconstruction plate was applied to the caudal aspect of the elbow for additional construct strength. Surgical evaluation was performed 8 weeks post-revision. Moderate mechanical lameness was reported with appropriate fusion of the EA site on radiographs. The ESF was removed at that time. On long term follow up, the owner reported that the dog had moderate lameness. The owner reported a good quality of life and excellent satisfaction with the outcome (Fig 3).

## Discussion

This is the first report discussing the major and minor complications and outcomes associated with EA using a variety of fixation types. While complication rates were high for all fixation types, complete elbow arthrodesis was radiographically achieved in 19/22 cases in this study. On long term follow up, 7/14 cases maintained mild to moderate lameness, 2/14 cases had no lameness, and 5/14 cases had severe non weight bearing lameness. While persistent lameness is anticipated long-term, lameness had improved in 9/14 cases. Irrespective of lameness,13/14 owners reported good to normal quality of life for their pets. These outcomes show that elbow arthrodesis, while under reported, remains a viable option for cases where other surgical repair options or amputation are not suitable.

In two previous publications, complication rates for EA were reported between 16–50% for both minor and major complications [10, 11]. In total, 19 complications, 7 minor and 12

major, are reported in the current study, which is consistent with the previously described complication rates for EA [10, 11]. Complications associated with ESF constructs were consistent with previous reports including pin migration and pin fracture resulting in surgical revision in all cases [8, 17–19]. The underlying cause for the complications is suspected to be multifactorial in nature including lack of standardized surgical repair and post-operative care, concurrent/underlying orthopedic disease, varying timeframes for veterinary clinical assessment, and owner compliance. Further studies are required to elucidate potential risk factors associated with the development of concurrent major and minor complications regardless of fixation style.

In the current study, the most common post-operative complication within each group was superficial SSI with an overall rate of 59%. SSI (superficial or deep) was not reported as a complication in either of the two previously published EA reports [10, 11]. SSI was reported in all groups of EA with a rate of 67% in the DCP group, 44% in LCP group, and 75% in the ESF group. These rates are higher than others that have been previously reported [20–22]. The cause of these higher infection rates is unknown but is suspected to be multifactorial in nature including the small sample size. Reported SSI rates in clean orthopedic procedures ranges from 0.6% to 7.1% [20, 23, 24]. However, a recent study regarding antimicrobial prophylaxis in orthopedic surgeries identified that SSI rates were highest in arthrodesis cases with a rate of 25% [24]. Post-operative infections were also commonly reported in ESF applications, with a post-operative infection rate of 39.2% in a recent report [17]. It was noted that the infection rate was dependent on the region of ESF application with superficial pin tract infection more common in the femur, humerus, radius and ulna, and pes [8, 17–19]. According to this study, the reported ESF infection rate may therefore be attributed to the region of application of the elbow.

The optimal use of antimicrobials in prevention of SSI is ongoing. Many studies suggest post-operative administration of antimicrobials in orthopedic procedures is protective against SSI in dogs [25–28]; however, there is evidence that prophylactic antibiotic administration in clean orthopedic procedures is not warranted [26]. Due to the retrospective nature of the present study, documentation of antimicrobial administration was not readily available in all cases but due to the high SSI rate, antibiotic administration along with culture and sensitivity testing should be considered in EA cases.

Mechanical lameness is an expected outcome of EA due to the fixed elbow joint angle [9–11]. In cases available for follow up, owners reported that in 7/14 cases, the dogs were lame with some degree of circumduction during ambulation. Due to the nature of the surgery, mechanical lameness is anticipated; however long-term physical examination would be required to confirm that the lameness was mechanical in nature. There are several published recommendations for elbow angle fixation, but an ideal angle of fixation has not been described. The currently recommended range of 110–159 degrees was strived for at the time of surgery [9–11, 13–16]. The mean angle of fixation was 120.9 degrees (range 93.5–140 degrees) which was considered appropriate based on the given range. On follow up, the case with the most acute angle of fixation (93.5 degrees) exhibited mechanical lameness but improved limb function overall, providing evidence that the clinical effects of these fixed angles remain largely unknown.

Surgical approach of the elbow can be performed either caudally or laterally. Appropriate cartilage debridement through a medial approach has been reported previously and was performed in a single case in this study [11, 29, 30]. The medial approach does not require an olecranon osteotomy and thus alleviates the risk of olecranon implantation failure or fragment displacement which occurred in three caudally approached cases in this study.

Follow-up was available for 14 cases but was not conducted at standardized post-operative times. Follow up questionnaires demonstrated that 13/14 dogs had a positive quality of life following EA. Owners reported no lameness in 2/14 cases, mild to moderate lameness in 7/14 cases, and severe lameness in 5/14 cases. Long term satisfaction was reported as good to excellent in 11/14 of owners. The responses to questions regarding dog functionality, quality of life and owner satisfaction with the outcome were interesting in the case with bilateral EA. On follow up, the staged bilateral EA dog (case #7, #8) had persistent non-weight bearing lameness bilaterally. The dog was placed in a mobility cart for thoracic limb orthopedic disease and a wheelchair for the pelvic limb orthopedic disease. Irrespective of the dog's lameness, the owner reported a good quality of life and good satisfaction with the outcome. This paradox between satisfied owners versus failure to achieve a functional outcome is most likely due to a multitude of subjective factors that have been put in place independently by each individual; however, according to Cook, et. al. it is possible to have satisfied owner yet an unacceptable outcome [12].

There are limitations of this study including its retrospective nature, limited heterogenous sample sizes, varying time frame to follow-up, lack of long-term surgical evaluation, and lack of consistent surgical/post-operative management. As with many retrospective studies, enrolment relied on submission of cases several years old, often with incomplete/inaccurate records, and follow-up was not always possible. The wide variability in institutional record-keeping precluded determination of prevalence of complications and potential risk factors associated with EA.

The current study shows that EA can be performed successfully; however, it has a high post-operative complication rate regardless of the method of fixation. The results of this study also indicate that EA using locking or non-locking plate systems provided acceptable restoration of limb function and pain alleviation. The majority of cases in this study had a good quality of life but maintained some degree of lameness and disuse at either a walk or a run in all groups. The ESF group had the highest rate of complications with 2/3 dogs having severe lameness on long term follow up resulting in unacceptable outcomes. Regardless, the majority of owners reported good to excellent satisfaction in all groups. SSI was the most commonly reported complications but has not been described previously as a complication in EA. Postoperative complications were identified in all groups of fixations; however, further research is needed to determine statistical inference between the groups and to identify potential risk factors associated with complications and EA fixation type.

## Acknowledgments

The authors would like to thank Dr. Michael Conzemius for his generous contribution of cases to this study.

## Author Contributions

**Conceptualization:** Elaine V. Dinwiddie, Albert C. Lynch, Brianna Miniter, Ron Ben-Amotz.

**Data curation:** Elaine V. Dinwiddie, Stan Veytsman, Guillaume Ragetly, Ron Ben-Amotz.

**Formal analysis:** Aaron Rendahl.

**Investigation:** Elaine V. Dinwiddie, Aaron Rendahl, Stan Veytsman, Ron Ben-Amotz.

**Methodology:** Guillaume Ragetly, Albert C. Lynch.

**Project administration:** Ron Ben-Amotz.

**Resources:** Aaron Rendahl, Stan Veytsman, Brianna Miniter, Ron Ben-Amotz.

**Software:** Aaron Rendahl.

**Supervision:** Ron Ben-Amotz.

**Validation:** Aaron Rendahl, Brianna Miniter.

**Visualization:** Albert C. Lynch, Ron Ben-Amotz.

**Writing – original draft:** Elaine V. Dinwiddie, Ron Ben-Amotz.

**Writing – review & editing:** Stan Veytsman, Guillaume Ragetly, Albert C. Lynch, Brianna Miniter, Ron Ben-Amotz.

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
