## [Decision Letter · Decision Letter 0]

4 Sep 2020

PONE-D-20-20573

Retrospective evaluation of post-operative complications of elbow arthrodesis (EA) in dogs: 22 cases (2009-2019)

PLOS ONE

Dear Dr. Dinwiddie,

Thank you for submitting your manuscript to PLOS ONE. After careful consideration, we feel that it has merit but does not fully meet PLOS ONE’s publication criteria as it currently stands. Therefore, we invite you to submit a revised version of the manuscript that addresses the points raised during the review process.

Many thanks for submitting your manuscript to PLOS One

Your manuscript was reviewed by two experts in the field, and they have suggested some revisions be made to it prior to acceptance.

If you could make these revisions, and write a response to reviewers, it will greatly expedite revision upon resubmission

I wish you the best of luck with your revisions

Hope you are keeping safe and well in these difficult times

Thanks

Simon

We look forward to receiving your revised manuscript.

Kind regards,

Simon Clegg, PhD

Academic Editor

PLOS ONE

Reviewers' comments:

Reviewer's Responses to Questions

**Comments to the Author**

1. Is the manuscript technically sound, and do the data support the conclusions?

Reviewer #1: No

Reviewer #2: Partly

2. Has the statistical analysis been performed appropriately and rigorously? 

Reviewer #1: N/A

Reviewer #2: Yes

3. Have the authors made all data underlying the findings in their manuscript fully available?

Reviewer #1: No

Reviewer #2: Yes

4. Is the manuscript presented in an intelligible fashion and written in standard English?

Reviewer #1: No

Reviewer #2: Yes

5. Review Comments to the Author

Reviewer #1: This case series reports the outcomes of elbow arthrodesis in 22 dogs. It is the largest series of its kind, and the manuscript also provides an update on the techniques as the previous largest case series was published decades ago. The authors are commended for collating the cases at multiple institutions, as well as attempting to obtain long term (albeit verbal) follow-up. There is potentially publishable material here, but several major flaws must be addressed:

- While the content of the manuscript was generally straight-forward, the scientific writing style needs much improvement, particularly pertaining to grammar. The authors should consider having the paper professionally edited prior to the next submission.

- There are many problems with sentence structure (too numerous to highlight every one), and anthropomorphisms (e.g. a group of dogs cannot report complications lines 36,179,183...).

- Additionally, it is not appropriate to report 'rates', 'common/uncommon' occurrences, and '%' when case numbers are so low (e.g. line 226 the most common major complication was pin migration (2/4)). Simply report the numbers as they are.

- There are over-reaching and misleading conclusions, which could be attributed to the lack of relevant data. Most importantly: reporting that most dogs had 'good to excellent quality of life' and 'high satisfaction' as one of the most relevant findings is highly misleading for this population. Indeed, most of the dogs had moderate or severe lameness on long term follow-up. The most striking case in point: (line 264-265)- one dog required a mobility cart due to lack of any weight-bearing on the arthrodesis limbs, yet the owners report good satisfaction and quality of life from the arthrodesis!! The authors should consider classifying outcomes as "full", "acceptable", "unacceptable" (per Cook et al., Vet Surg 2010) based on the questionnaire results. For instance, this reviewer would suggest outcomes of dogs with 'moderate' or 'severe' overall limb disability would be 'unacceptable'. Note that, per Cook et al Vet Surg 2010, it is possible to have a satisfied owner yet an unacceptable outcome. On a similar note- how is it possible to diagnose a 'mechanical' lameness by telephone conversation?

- The case numbers within groups are too low to draw any major conclusions from comparisons between groups - e.g. only 3 dogs with ESF had follow-up. Therefore, 'recommendations' (e.g. cannot recommend ESF) should be tempered.

- While the manuscript title suggests the case series focuses on complications, there is a clear intent to report overall outcomes as well. Therefore, 1) the title must be adjusted, and 2) more information regarding outcomes should be provided; namely- (when) was union of the arthrodesis documented for each case?

Specific comments:

Line 43: Conclusions: major complication rate was high for all groups. Why single out ESF if final result not obviously different?

Line 51-53: not sure how this is a ‘paradigm’; suggest deleting this sentence.

Paragraph starting line 65: Please report how many cases of elbow arthrodesis have been previously described. This will give better context for the relevance of this manuscript.

Line 86: why exclude these methods?

Line 120: What were the outcomes of the non-grafted cases? Grafting is a fundamentally important aspect of arthrodesis

Line 137-138: Provide the scales for satisfaction (excellent, good, fair, poor)

Line 154: please ensure none of the cases have been described in other papers; or otherwise disclose the information

Line 159-165: Redundant statements (already mentioned in the methods); delete

Line 172: Carpal valgus??

Line 178-186: all redundant information as these values are provided later in the results again

Table 2 is not easy to read; consider revising to chart form.

Line 244 vs line 253: range is 1-9 years yet 2 dogs were euthanized < 1 year postoperatively?

Line 285-288: cut and pasted out of results; please rephrase

Line 291-295: it is hard to agree with the logic here based on the data provided (i.e. that minor complications will lead to major complications).

Paragraph starting line 296: this paragraph should discuss SSI occurrence for all groups, yet only ESF is discussed.

Line 311: How is it known that the lameness was mechanical, since outcomes were defined by telephone conversations with owners?. There is no mention of circumduction in the results?

Paragraph starting line 321: Is this supposed to be the limitations paragraph? There are many more limitations than the variable approaches taken i.e. incomplete/inaccurate records, lack of veterinarian assessed follow-up, highly heterogenous population etc etc.

Paragraph starting line 333: The arguments here are based on extremely weak data. Indeed, from the table, it is not obvious that older animals had more complications or poorer outcomes.

Line 348: If you are recommending a validated assessment, why was one not used??

Line 258: "Good use of the operated limb" in the 'majority of patients' is simply not true; 10/14 dogs had moderate or severe limb disability; only 2/14 dogs did not have lameness at walk and run

Line 359-360: Where in the results is this shown? i.e. that ESF group had lacked improvement post-operatively and had lowest satisfaction (2 out of 3 owners were actually satisfied)?

Please consider adding a paragraph in the discussion on the paradox between satisfied owners vs lack of good limb use. One possibility is that the owners were simply happy that their dog was not amputated.

Reviewer #2: I think that this study has potential for publication as it is a nice series of reports on elbow arthrodesis in dogs, which is an under reported area, and this is a nice and fairly large study for this area. It is nice that it is multi centred too and the authors deserve a lot of credit for it. It is a nice study, and a nice paper with some good results, but I think that some are unclear and in some places, over interpreted.

I struggled to follow certain parts of it as it was quite contradictory. For example, how can dogs have an excellent quality of life, yet be moderate to severely lame? Including one which had a mobility cart? The two seem somewhat contradictory to me. I think I understand what you mean, but I think it needs some quantification.

I think I would also be tempted to edge on the side of caution with your conclusions, as although numbers are relatively large for this field, overall they are quite small.

The title is also a little misleading (wrong word but couldn’t think up a better one). You have a lot on clinical final outcomes, so that maybe better in the title too?

Line 54-57- its not the patients which have the difficult choices to make- it’s the owners!

Line 66- up to 100% means very little, can you be more specific?

Line 85- how was quality of life assessed?

Line 86- why were these methods excluded? Is there a specific reason?

Line 106- . In total, twenty-two EA procedures were included in this study (Add in comma)

Line 120- can you mention the outcomes of the cases? Particularly the grafts as these can sometimes be tricky to take

Line 135-139- these are quite open, subjective categories to ask. Were there scales used? If so, please include them

Some of the results are included already above, please consider revising

I struggled to follow table 2- this may be easier as a figure, or with more explanation.

Line 255- again it would be nice to know how the quality of life was judged

Line 285-288 is better in results, and used here to discuss prevalence etc

Line 291- is a major complaint not just a progression of a minor one? I can see what you are saying here but its unclear. Perhaps a major complaint masks a minor?

Line 303- To the authors knowledge, there is no published infection rate of (add in comma)

Line 311- hpw do you know that it is mechanical lameness? Unlikely that the owner would pick this up.

Line 321- the limitations are far greater than those explained, including inaccurate records, different vets, poor follow up etc

Line 333- I have concerns about some of the arguments here as the data isn’t that strong- maybe have a read over it and consider rewording it?

Line 358- good use of the limb seems in direct opposition to some of what you have said previously?

Line 359- isn’t the ESF group the lowest improvement group post op?

6. PLOS authors have the option to publish the peer review history of their article (what does this mean?). If published, this will include your full peer review and any attached files.

Reviewer #2: No

---

## [Author Response · Author response to Decision Letter 0]

17 Jun 2021

Response to reviewers:

Thank you both for taking the time to review, and comment about our manuscript. 

We hope that with this version all comments were addressed. Please let us know if any additional changes are needed. 

Sincerely 

Dr. Dinwiddie and authors 

Reviewer #1: 

 While the content of the manuscript was generally straight-forward, the scientific writing style needs much improvement, particularly pertaining to grammar. The authors should consider having the paper professionally edited prior to the next submission.

-The manuscript has been re-written and professionally edited by experienced authors multiple times prior to re-submission making direct correction to grammar, sentence structure, and writing style. 

Additionally, it is not appropriate to report 'rates', 'common/uncommon' occurrences, and '%' when case numbers are so low (e.g. line 226 the most common major complication was pin migration (2/4)). Simply report the numbers as they are.

-The sentence structures of the manuscript have been addressed to reflect numbers simply as reported. There are only 2 previously reported articles regarding EA which have are referenced in the manuscript. This lack of information regarding EA is noted starting in line 66

There are over-reaching and misleading conclusions, which could be attributed to the lack of relevant data. Most importantly: reporting that most dogs had 'good to excellent quality of life' and 'high satisfaction' as one of the most relevant findings is highly misleading for this population. Indeed, most of the dogs had moderate or severe lameness on long term follow-up. The most striking case in point: (line 264-265)- one dog required a mobility cart due to lack of any weight-bearing on the arthrodesis limbs, yet the owners report good satisfaction and quality of life from the arthrodesis!! The authors should consider classifying outcomes as "full", "acceptable", "unacceptable" (per Cook et al., Vet Surg 2010) based on the questionnaire results. For instance, this reviewer would suggest outcomes of dogs with 'moderate' or 'severe' overall limb disability would be 'unacceptable'. Note that, per Cook et al Vet Surg 2010, it is possible to have a satisfied owner yet an unacceptable outcome. On a similar note- how is it possible to diagnose a 'mechanical' lameness by telephone conversation?

-The conclusions have been edited to reflect that while arthrodesis can be achieved via multiple fixation methods, care should be taken in performing this procedure because all fixation groups had complications. Over-reaching statements have been removed and/or edited to reflect that. In addition, we have added statements discussing Cook et al’s paper reflecting that there was case with satisfied owners yet unacceptable outcome (lines 395-399). Statements were also added to discuss mechanical lameness at 8 week surgical rechecks (lines 202-208) and that while mechanical lameness is expected, without physical exams, it is not possible to confirm that the lameness is indeed mechanical in nature (lines 355-367)

The case numbers within groups are too low to draw any major conclusions from comparisons between groups - e.g. only 3 dogs with ESF had follow-up. Therefore, 'recommendations' (e.g. cannot recommend ESF) should be tempered.

-Statements singling out any one type of fixation have been removed as all types of fixations had complications. 

While the manuscript title suggests the case series focuses on complications, there is a clear intent to report overall outcomes as well. Therefore, 1) the title must be adjusted, and 2) more information regarding outcomes should be provided; namely- (when) was union of the arthrodesis documented for each case?

 -Title has been edited Evaluation of post-operative complications, outcome, and long-term owner satisfaction of elbow arthrodesis (EA) in dogs: 22 cases (2009-2019)

In addition, we have added dialogue regarding post-operative evaluations as well as information regarding arthrodesis union (lines 161-162)

Line 43: Conclusions: major complication rate was high for all groups. Why single out ESF if final result not obviously different?

-Statements regarding conclusions have been edited to reflect that while complication rates were high for all fixation types, arthrodesis is achievable in all fixation types as well (lines 46-47)

Line 51-53: not sure how this is a ‘paradigm’; suggest deleting this sentence.

-sentence deleted

Paragraph starting line 65: Please report how many cases of elbow arthrodesis have been previously described. This will give better context for the relevance of this manuscript.

-Paragraph starting line 63 discusses infrequency of EA reporting to give relevance of this manuscript (One paper written in the last 10 years)

Line 86: why exclude these methods?

-the methods of fixation that the authors wished to include in this study were bone plate and ESF which are the most common. Any other form was eliminated for continuity purposes and so that the categories of EA could be maintained. 

-Cases included were those with EA performed by plate fixation or ESF and reflected in the abstract lines 26-30

Line 120: What were the outcomes of the non-grafted cases? Grafting is a fundamentally important aspect of arthrodesis

 -Paragraph starting line 163 discussed outcome of non-grafted cases

Line 137-138: Provide the scales for satisfaction (excellent, good, fair, poor)

 -Scales have been added for all evaluations (surgical and owner) in lines 92-103

Line 154: please ensure none of the cases have been described in other papers; or otherwise disclose the information

 -None of these cases have been previously reported. The other 2 papers regarding EA are referenced in the manuscript

Line 159-165: Redundant statements (already mentioned in the methods); delete

 -Redundant information and statement have been removed and/or edited 

Table 2 is not easy to read; consider revising to chart form.

 -Table 2 was revised into individual tables (Table 2, 3, and 4) to reflect responses to individual fixation types. The owners responses to the question on the left hand is written next to the possible options in parentheses to the right

Line 244 vs line 253: range is 1-9 years yet 2 dogs were euthanized < 1 year postoperatively?

 -range was updated to reflect correct time frame (7 months-9 years)

Line 285-288: cut and pasted out of results; please rephrase

 -redundant information was removed and results section restated

Line 291-295: it is hard to agree with the logic here based on the data provided (i.e. that minor complications will lead to major complications).

 -Paragraph has been re-written to discuss infection associated with these fixations as we cannot conclude that one minor complication leads to a major. Discussion thus surrounds the SSI portion of complications in paragraph starting 331 

Paragraph starting line 296: this paragraph should discuss SSI occurrence for all groups, yet only ESF is discussed.

 -Paragraph starting 331 edited to discusses SSI. In all forms of EA fixation, not just ESF

Line 311: How is it known that the lameness was mechanical, since outcomes were defined by telephone conversations with owners?. There is no mention of circumduction in the results?

-Information regarding surgical evaluation at 8 weeks post operatively has been added to include mechanical lameness as well as circumduction (lines 63-68). Mechanical lameness is expected with this procedure; however, without performing PE, we cannot confirm that on long term evaluations with the owners via phone call if the lamenesses were indeed mechanical. This is reflected in discussion starting lines 355-360. 

Paragraph starting line 321: Is this supposed to be the limitations paragraph? There are many more limitations than the variable approaches taken i.e. incomplete/inaccurate records, lack of veterinarian assessed follow-up, highly heterogenous population etc etc.

 -Limitations paragraph has been re-written to reflect a more conclusive list starting line 406

Paragraph starting line 333: The arguments here are based on extremely weak data. Indeed, from the table, it is not obvious that older animals had more complications or poorer outcomes.

-Paragraph was removed as argument was weak and not founded on information gathered for this manuscript

Line 348: If you are recommending a validated assessment, why was one not used??

- We used a modified version of the previously reported questionnaire as a means to keep data consistent. The other validating system (i.e. force plate analysis) would be useful post operatively, but due to the retrospective nature of this study, we could not conduct that type of analysis. 

Line 258: "Good use of the operated limb" in the 'majority of patients' is simply not true; 10/14 dogs had moderate or severe limb disability; only 2/14 dogs did not have lameness at walk and run

-10/14 dogs had mild to moderate lameness long term which is an improvement for these patients from pre-operatively. Again, mechanical lameness is expected so we anticipate lameness with some of these cases. This should be more evident now that the Table 2 has been broken down into individual tables to reflect the long term outcome of patients. 

Line 359-360: Where in the results is this shown? i.e. that ESF group had lacked improvement post-operatively and had lowest satisfaction (2 out of 3 owners were actually satisfied)?

-ESF group did have a good outcome which is now reflected in the ESF complications and outcome section begin line 257. Outcome for each fixation type has been re-written and added to the respective section 

Please consider adding a paragraph in the discussion on the paradox between satisfied owners vs lack of good limb use. One possibility is that the owners were simply happy that their dog was not amputated.

-Paragraph starting 383 was added to discuss this paradox

Reviewer #2: 

I struggled to follow certain parts of it as it was quite contradictory. For example, how can dogs have an excellent quality of life, yet be moderate to severely lame? Including one which had a mobility cart? The two seem somewhat contradictory to me. I think I understand what you mean, but I think it needs some quantification.

-Paragraph starting line 383 was added to reflect that there can be pleased owners that perceive a good quality of life for their pet despite that they have an unacceptable outcome. 

I think I would also be tempted to edge on the side of caution with your conclusions, as although numbers are relatively large for this field, overall they are quite small.

 -Conclusions have been tempered and re-written to reflect the data given here, that all fixation types can result in complete arthrodesis but all groups had complications; therefore, the conclusions are merely cautionary in nature about EA as a procedure regardless of fixation type

The title is also a little misleading (wrong word but couldn’t think up a better one). You have a lot on clinical final outcomes, so that maybe better in the title too?

- Title has been edited Evaluation of post-operative complications, outcome, and long-term owner satisfaction of elbow arthrodesis (EA) in dogs: 22 cases (2009-2019)

Line 54-57- its not the patients which have the difficult choices to make- it’s the owners!

 -The Abstract and Introduction have both been revised and include discussion of treatment of cases with end stage elbow disease and what options are available for owners. 

Line 85- how was quality of life assessed?

-Quality of life was one of the long term questionnaire questions and a subjective measurement by the owners meant for them to assess how their pet was doing at that point long term (Table 2, 3, 4). 

Line 86- why were these methods excluded? Is there a specific reason?

-The line was re-written- the cases included were cases that had EA performed by either LCP, DCP, or ESF. 

Line 106- . In total, twenty-two EA procedures were included in this study (Add in comma)

-The manuscript was professionally edited again and hopefully addressed most of the grammatical errors. This sentence was re-written with the comma added

Line 120- can you mention the outcomes of the cases? Particularly the grafts as these can sometimes be tricky to take

-Case outcomes have been included in the paragraph regarding each respective fixation type (i.e. DCP complications and outcome). Table 2 had been broken down to reflect each fixation type, so are now Table 2, Table 3, Table 4. In addition, images of each fixation type have been included as well. 

Some of the results are included already above, please consider revising

-The materials/methods and results sections have been revised to better include information in correct sections. 

I struggled to follow table 2- this may be easier as a figure, or with more explanation.

-Table 2 was broken down to reflect owner responses to individual fixation types and are now Table 2, Table 3, Table 4

Line 291- is a major complaint not just a progression of a minor one? I can see what you are saying here but its unclear. Perhaps a major complaint masks a minor?

-Both major and minor complications have been revised in both Table 1 as well as the text to better clarify what is minor vs major. 

Line 303- To the authors knowledge, there is no published infection rate of (add in comma)

-This paragraph has been edited to discuss infection in all types of fixation not only ESF in paragraph beginning 331

Line 311- hpw do you know that it is mechanical lameness? Unlikely that the owner would pick this up.

- Statements were added to discuss mechanical lameness at 8 week surgical rechecks (lines 202-208) and that while mechanical lameness is expected, without physical exams, it is not possible to confirm that the lameness is indeed mechanical in nature (lines 355-367)

Line 321- the limitations are far greater than those explained, including inaccurate records, different vets, poor follow up etc

-The paragraph regarding limitations has been edited and includes more limitations than previously discussed. This is discussed in paragraph starting at line 406

Line 333- I have concerns about some of the arguments here as the data isn’t that strong- maybe have a read over it and consider rewording it?

-The discussion portion was re-written to take into consideration that due to the low numbers of cases, some overarching conclusions couldn’t be made. This paragraph was reworded with specific changes to the arguments. 

Line 359- isn’t the ESF group the lowest improvement group post op?

-The ESF group had the most complications associated with it but only 1 patient had low improvement. This section was re-written to reflect complications and outcome for ESF lines 258-294

---

## [Decision Letter · Decision Letter 1]

6 Jul 2021

PONE-D-20-20573R1

Evaluation of post-operative complications, outcome, and long-term owner satisfaction of elbow arthrodesis (EA) in 22 dogs

PLOS ONE

Dear Dr. Ben-Amotz,

Thank you for submitting your manuscript to PLOS ONE. After careful consideration, we feel that it has merit but does not fully meet PLOS ONE’s publication criteria as it currently stands. Therefore, we invite you to submit a revised version of the manuscript that addresses the points raised during the review process.

Many thanks for submitting your manuscript to PLOS One

It was reviewed by two experts in the field as reviewed the original submission, and they have recommended some further modifications be made prior to acceptance

I therefore invite you to make these changes and to write a response to reviewers which will expedite revision upon resubmission

I wish you the best of luck with your modifications

Hope you are keeping safe and well in these difficult times

Thanks

Simon

We look forward to receiving your revised manuscript.

Kind regards,

Simon Clegg, PhD

Academic Editor

PLOS ONE

Journal Requirements:

Reviewers' comments:

Reviewer's Responses to Questions

**Comments to the Author**

1. If the authors have adequately addressed your comments raised in a previous round of review and you feel that this manuscript is now acceptable for publication, you may indicate that here to bypass the “Comments to the Author” section, enter your conflict of interest statement in the “Confidential to Editor” section, and submit your "Accept" recommendation.

Reviewer #1: (No Response)

Reviewer #2: All comments have been addressed

2. Is the manuscript technically sound, and do the data support the conclusions?

Reviewer #1: Partly

Reviewer #2: Yes

3. Has the statistical analysis been performed appropriately and rigorously? 

Reviewer #1: No

Reviewer #2: Yes

4. Have the authors made all data underlying the findings in their manuscript fully available?

Reviewer #1: Yes

Reviewer #2: Yes

5. Is the manuscript presented in an intelligible fashion and written in standard English?

Reviewer #1: Yes

Reviewer #2: Yes

6. Review Comments to the Author

Reviewer #1: Thank you for the revisions, the manuscript is improved.

General comments:

- Do not use the word ‘patient’ unless referring to human subject. Use ‘dog’ or ‘case’ instead.

- Delete all % values- the numbers are too small in this study.

- The results and discussion are very lengthy and could be condensed further.

Specific comments:

Line 36,37: “Complications were reported in X/X, Y/Y, and Z/Z for the DCP, LCP, and ESF groups, respectively”

Do not jumble methods with results in abstract – i.e. move sentence on line 37 to methods area

Abstract conclusion: Yes it can be done, but persistent lameness is expected and complication rate is high.

Line 54: amputation is a radical surgical intervention – delete ‘amputation’

Line 63: Do not begin sentence with acronym

Line 64: what is meant by ‘the most recent literature’?

Line 70: Something is missing between the last two paragraphs of the introduction. Need to provide some comment about advent of new implants i.e. locking plates that might influence success rates.

Line 78: acronym not used for EA

Line 93: Outcome was based on evaluations/radiographs at what minimum time frame?

Line 97: the time range provided belongs in the results.

Line 134: why 9/9? The number of cases is already listed above

Tables 2-4 should be combined

Discussion:

First paragraph: Need to explicitly state that, while persistent lameness was typical, lameness had improved.

Line 333: Do not report % - numbers are too small therefore % are misleading.

Line 359: Was mechanical in nature

Line 375: A little too speculative; consider deleting this paragraph (manuscript is already very long)

Line 391-393: this needs to be stated in the results

Line 398-404: Consider deleting from “To increase the accuracy…” to the end of the paragraph.

Reviewer #2: I wish to thank the authors for making the previously suggested modifications. I have read the manuscript, and although it reads well, I still have a few minor comments to address prior to acceptance. These are listed below, but I don’t expect to see the manuscript again prior to acceptance so I want to wish you all the best with your future research.

Please note that these are only suggestions and I shall not be disappointed if they are not actioned.

Within the abstract, is it possible to have both the n and a % value? So lines 40. 41 and 42

Line 110- there is only 21 here, not the 22 which were included, Can you please add in the 22nd?

Line 116- again there are only 20 here, can you add in the other two please?

Line 143- completed, the bone graft (reword)

Line 144- grafted, an autogenous …(reword)

Line 153- screws rather than screw

Line 165- did not receive a bone graft (reword)

Line 179- non union, and deep SSI (reword)

Line 228- screw, and humeral fissure (reword)

7. PLOS authors have the option to publish the peer review history of their article (what does this mean?). If published, this will include your full peer review and any attached files.

Reviewer #2: No

---

## [Author Response · Author response to Decision Letter 1]

12 Jul 2021

Response to reviewers:

Thank you both again for taking the time to review, and comment on our manuscript. 

We hope that with this version all comments were addressed. This manuscript is required to obtain ACVS credentials by August 1, 2021. Please let us know if any additional changes are needed prior to acceptance. 

Sincerely, 

Dr. Dinwiddie and authors 

Reviewer #1: Thank you for the revisions, the manuscript is improved.

General comments:

Do not use the word ‘patient’ unless referring to human subject. Use ‘dog’ or ‘case’ instead.

-The term ‘patient’ was replaced with either dog or case throughout the manuscript

Delete all % values- the numbers are too small in this study.

 -Percentage values were removed from the manuscript with the exception of the discussion of infection rates for this study. The statement of “small sample size” was added to the discussion paragraph to specifically recall that sample size plays a factor in these infection rates

The results and discussion are very lengthy and could be condensed further.

 -Both the results and discussion have had portions removed or revised to condense the material. 

Specific comments:

Line 36,37: “Complications were reported in X/X, Y/Y, and Z/Z for the DCP, LCP, and ESF groups, respectively”

-This line has been reworded as suggested above

Do not jumble methods with results in abstract – i.e. move sentence on line 37 to methods area

-Sentence removed from abstract and stated within methods section

Abstract conclusion: Yes it can be done, but persistent lameness is expected and complication rate is high.

-The concluding sentences were combined to read better 

Line 54: amputation is a radical surgical intervention – delete ‘amputation’

-‘amputation’ was deleted

Line 63: Do not begin sentence with acronym

-Changed EA to Elbow Arthrodesis

Line 64: what is meant by ‘the most recent literature’?

 -Defined and edited to say ‘in the past 10 years of literature’

Line 70: Something is missing between the last two paragraphs of the introduction. Need to provide some comment about advent of new implants i.e. locking plates that might influence success rates.

 -Added paragraph addressing available implants

Line 78: acronym not used for EA

-Changed elbow arthrodesis to ‘EA’

Line 93: Outcome was based on evaluations/radiographs at what minimum time frame?

-Added ‘average of 9 weeks post operatively’. The full timeframe range is stated in the results section

Line 97: the time range provided belongs in the results.

-Moved timeframe to results section

Line 134: why 9/9? The number of cases is already listed above

-Removed ‘9/9’ and replaced with ‘all cases’

Tables 2-4 should be combined

 -The tables have been combined into a single Table 2 and references within the manuscript edited to reflect the change

Discussion:

First paragraph: Need to explicitly state that, while persistent lameness was typical, lameness had improved.

-Added sentence to state that lameness had improved in 9/14 cases 

Line 333: Do not report % - numbers are too small therefore % are misleading.

-Percentages have been removed

Line 359: Was mechanical in nature

-edited sentence to say ‘was mechanical in nature’

Line 375: A little too speculative; consider deleting this paragraph (manuscript is already very long)

-paragraph was removed

Line 391-393: this needs to be stated in the results

-This sentence was moved to the results and reworded to be included in the Discussion

Line 398-404: Consider deleting from “To increase the accuracy…” to the end of the paragraph.

-This section has been removed along with the references that were stated in the section

Reviewer #2: 

I wish to thank the authors for making the previously suggested modifications. I have read the manuscript, and although it reads well, I still have a few minor comments to address prior to acceptance. These are listed below, but I don’t expect to see the manuscript again prior to acceptance so I want to wish you all the best with your future research.

Please note that these are only suggestions and I shall not be disappointed if they are not actioned.

Within the abstract, is it possible to have both the n and a % value? So lines 40. 41 and 42

-The percentages have been removed

Line 110- there is only 21 here, not the 22 which were included, Can you please add in the 22nd?

-This sentence is correct in that there were only 21 dogs- but one case had bilateral elbows operated making it 22 total procedures

Line 116- again there are only 20 here, can you add in the other two please?

-Recounted all patients and appropriately categorized. Number is now 22 including the bilateral case

Line 143- completed, the bone graft (reword)

-Did not add the word ‘the’

Line 144- grafted, an autogenous …(reword)

-Added the word ‘an’

Line 153- screws rather than screw

-Edited to make word plural

Line 165- did not receive a bone graft (reword)

-Added ‘a’

Line 179- non union, and deep SSI (reword)

-Added ‘and’

Line 228- screw, and humeral fissure (reword)

 -Added ‘and’

---

## [Editor Report · Decision Letter 2]

16 Jul 2021

Evaluation of post-operative complications, outcome, and long-term owner satisfaction of elbow arthrodesis (EA) in 22 dogs

PONE-D-20-20573R2

Dear Dr. Ben-Amotz,

We’re pleased to inform you that your manuscript has been judged scientifically suitable for publication and will be formally accepted for publication once it meets all outstanding technical requirements.

Kind regards,

Simon Clegg, PhD

Academic Editor

PLOS ONE

Additional Editor Comments:

Many thanks for resubmitting your manuscript to PLOS One

As you have addressed all the comments and the manuscript reads well, I have recommended it for publication

You should hear from the Editorial Office shortly.

It was a pleasure working with you and I wish you the best of luck for your future research

Hope you are keeping safe and well in these difficult times

Thanks

Simon

---

## [Editor Report · Acceptance letter]

21 Jul 2021

PONE-D-20-20573R2 

Evaluation of post-operative complications, outcome, and long-term owner satisfaction of elbow arthrodesis (EA) in 22 dogs 

Dear Dr. Ben-Amotz:

I'm pleased to inform you that your manuscript has been deemed suitable for publication in PLOS ONE. Congratulations! Your manuscript is now with our production department. 

Kind regards, 

on behalf of

Dr. Simon Clegg 

Academic Editor

PLOS ONE